# UNBIASING TRUNCATED BACKPROPAGATION THROUGH TIME

## ABSTRACT

*Truncated Backpropagation Through Time* (truncated BPTT, Jaeger (2005)) is a widespread method for learning recurrent computational graphs. Truncated BPTT keeps the computational benefits of *Backpropagation Through Time* (BPTT Werbos (1990)) while relieving the need for a complete backtrack through the whole data sequence at every step. However, truncation favors short-term dependencies: the gradient estimate of truncated BPTT is biased, so that it does not benefit from the convergence guarantees from stochastic gradient theory. We introduce *Anticipated Reweighted Truncated Backpropagation* (ARTBP), an algorithm that keeps the computational benefits of truncated BPTT, while providing unbiasedness. ARTBP works by using variable truncation lengths together with carefully chosen compensation factors in the backpropagation equation. We check the viability of ARTBP on two tasks. First, a simple synthetic task where careful balancing of temporal dependencies at different scales is needed: truncated BPTT displays unreliable performance, and in worst case scenarios, divergence, while ARTBP converges reliably. Second, on Penn Treebank character-level language modelling Mikolov et al. (2012), ARTBP slightly outperforms truncated BPTT.

*Backpropagation Through Time* (BPTT) Werbos (1990) is the de facto standard for training recurrent neural networks. However, BPTT has shortcomings when it comes to learning from very long sequences: learning a recurrent network with BPTT requires unfolding the network through time for as many timesteps as there are in the sequence. For long sequences this represents a heavy computational and memory load. This shortcoming is often overcome heuristically, by arbitrarily splitting the initial sequence into subsequences, and only backpropagating on the subsequences. The resulting algorithm is often referred to as *Truncated Backpropagation Through Time* (truncated BPTT, see for instance Jaeger (2005)). This comes at the cost of losing long term dependencies.

We introduce *Anticipated Reweighted Truncated BackPropagation* (ARTBP), a variation of truncated BPTT designed to provide an unbiased gradient estimate, accounting for long term dependencies. Like truncated BPTT, ARTBP splits the initial training sequence into subsequences, and only back-propagates on those subsequences. However, unlike truncated BPTT, ARTBP splits the training sequence into variable size subsequences, and suitably modifies the backpropagation equation to obtain unbiased gradients.

Unbiasedness of gradient estimates is the key property that provides convergence to a local optimum in stochastic gradient descent procedures. Stochastic gradient descent with biased estimates, such as the one provided by truncated BPTT, can lead to divergence even in simple situations and even with large truncation lengths (Fig. 3).

ARTBP is experimentally compared to truncated BPTT. On truncated BPTT failure cases, typically when balancing of temporal dependencies is key, ARTBP achieves reliable convergence thanks to unbiasedness. On small-scale but real world data, ARTBP slightly outperforms truncated BPTT on the test case we examined.

ARTBP formalizes the idea that, on a day-to-day basis, we can perform short term optimization, but must reflect on long-term effects once in a while; ARTBP turns this into a provably unbiased overall gradient estimate. Notably, the many short subsequences allow for quick adaptation to the data, while preserving overall balance.

# 1 RELATED WORK

BPTT Werbos (1990) and its truncated counterpart Jaeger (2005) are nearly uncontested in the recurrent learning field. Nevertheless, BPTT is hardly applicable to very long training sequences, as it requires storing and backpropagating through a network with as many layers as there are timesteps Sutskever (2013). Storage issues can be partially addressed as in Gruslys et al. (2016), but at an increased computational cost. Backpropagating through very long sequences also implies performing fewer gradient descent steps, which significantly slows down learning Sutskever (2013).

Truncated BPTT heuristically solves BPTT deficiencies by chopping the initial sequence into evenly sized subsequences. Truncated BPTT truncates gradient flows between contiguous subsequences, but maintains the recurrent hidden state of the network. Truncation biases gradients, removing any theoretical convergence guarantee. Intuitively, truncated BPTT has trouble learning dependencies above the range of truncation. [1]

*NoBackTrack* Ollivier et al. (2015) and *Unbiased Online Recurrent Optimization* (UORO) Tallec & Ollivier (2017) both scalably provide unbiased online recurrent learning algorithms. They take the more extreme point of view of requiring memorylessness, thus forbidding truncation schemes and any storage of past states. NoBackTrack and UORO's fully online, streaming structure comes at the price of noise injection into the gradient estimates via a random rank-one reduction. ARTBP's approach to unbiasedness is radically different: ARTBP is not memoryless but does not inject artificial noise into the gradients, instead, compensating for the truncations directly inside the backpropagation equation.

# 2 BACKGROUND ON RECURRENT MODELS

The goal of recurrent learning algorithms is to optimize a parametric dynamical system, so that its output sequence, or predictions, is as close as possible to some target sequence, known a priori. Formally, given a dynamical system with state $s$, inputs $x$, parameter $\theta$, and transition function $F$,

$$s_{t+1} = F(x_{t+1}, s_t, \theta) \tag{1}$$

the aim is to find a $\theta$ minimizing a total loss with respect to target outputs $o_t^*$ at each time,

$$\mathcal{L}_T = \sum_{t=1}^{T} \ell_t = \sum_{t=1}^{T} \ell(s_t, o_t^*). \tag{2}$$

A typical case is that of a standard recurrent neural network (RNN). In this case, $s_t = (o_t, h_t)$, where $o_t$ are the activations of the output layer (encoding the predictions), and $h_t$ are the activations of the hidden recurrent layer. For this simple RNN, the dynamical system takes the form

$$h_{t+1} = \tanh(W_x\, x_{t+1} + W_h\, h_t + b) \tag{3}$$
$$o_{t+1} = W_o h_{t+1} \tag{4}$$
$$\ell_{t+1} = \ell(o_{t+1}, o_{t+1}^*) \tag{5}$$

with parameters $\theta = (W_x, W_h, b)$.

Commonly, $\theta$ is optimized via a gradient descent procedure, i.e. iterating

$$\theta \leftarrow \theta - \eta \frac{\partial \mathcal{L}_T}{\partial \theta} \tag{6}$$

where $\eta$ is the learning rate. The focus is then to efficiently compute $\partial \mathcal{L}_T / \partial \theta$.

*Backpropagation through time* is a method of choice to perform this computation. BPTT computes the gradient by unfolding the dynamical system through time and backpropagating through it, with each timestep corresponding to a layer. BPTT decomposes the gradient as a sum, over timesteps $t$, of the effect of a change of parameter at time $t$ on all subsequent losses. Formally,

$$\frac{\partial \mathcal{L}_T}{\partial \theta} = \sum_{t=1}^{T} \delta\ell_t\, \frac{\partial F}{\partial \theta}(x_t, s_{t-1}, \theta) \tag{7}$$

---

[1] Still, as the hidden recurrent state is not reset between subsequences, it may contain hidden information about the distant past, which can be exploited Sutskever (2013).

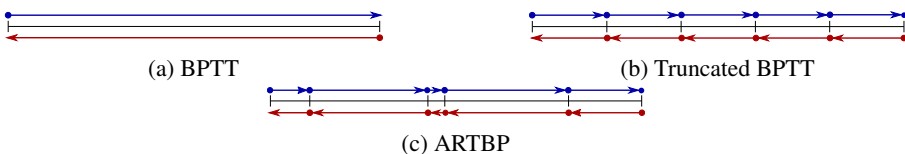

Figure 1: Graphical representation of BPTT, truncated BPTT and ARTBP. Blue arrows represent forward propagations, red arrows backpropagations. Dots represent either internal state resetting or gradient resetting.

where $\delta\ell_t := \frac{\partial \mathcal{L}_T}{\partial s_t}$ is computed backward iteratively according to the backpropagation equation

$$\begin{cases} \delta\ell_T = \frac{\partial\ell}{\partial s}(s_T, o_T^*) \\ \delta\ell_t = \delta\ell_{t+1}\frac{\partial F}{\partial s}(x_{t+1}, s_t, \theta) + \frac{\partial\ell}{\partial s}(s_t, o_t^*). \end{cases} \tag{8}$$

These backpropagation equations extend the classical ones Jaeger (2005), which deal with the case of a simple RNN for $F$.

Unfortunately, BPTT requires processing the full sequence both forward and backward. This requires maintaining the full unfolded network, or equivalently storing the full history of inputs and activations (though see Gruslys et al. (2016)). This is impractical when very long sequences are processed with large networks: processing the whole sequence at every gradient step slows down learning.

Practically, this is alleviated by truncating gradient flows after a fixed number of timesteps, or equivalently, splitting the input sequence into subsequences of fixed length, and only backpropagating through those subsequences. [2] This algorithm is referred to as *Truncated BPTT*. With truncation length $L < T$, the corresponding equations just drop the recurrent term $\delta\ell_{t+1}\frac{\partial F}{\partial s}(x_{t+1}, s_t, \theta)$ every $L$ time steps, namely,

$$\hat{\delta\ell}_t := \begin{cases} \frac{\partial\ell}{\partial s}(s_t, o_t^*) & \text{if } t \text{ is a multiple of } L \\ \hat{\delta\ell}_{t+1}\frac{\partial F}{\partial s}(x_{t+1}, s_t, \theta) + \frac{\partial\ell}{\partial s}(s_t, o_t^*) & \text{otherwise.} \end{cases} \tag{9}$$

This also allows for online application: for instance, the gradient estimate from the first subsequence $t = 1\ldots, L$ does not depend on anything at time $t > L$.

However, this gradient estimation scheme is heuristic and provides biased gradient estimates. In general the resulting gradient estimate can be quite far from the true gradient even with large truncations $L$ (Section 6). Undesired behavior, and, sometimes, divergence can follow when performing gradient descent with truncated BPTT (Fig. 3).

# 3 ANTICIPATED REWEIGHTED BACKPROPAGATION THROUGH TIME: UNBIASEDNESS THROUGH REWEIGHTED STOCHASTIC TRUNCATION LENGTHS

Like truncated BPTT, ARTBP splits the initial sequence into subsequences, and only performs backpropagation through time on subsequences. However, contrary to the latter, it does not split the sequence evenly. The length of each subsequence is sampled according to a specific probability distribution. Then the backpropagation equation is modified by introducing a suitable reweighting factor at every step to ensure unbiasedness. Figure 1 demonstrates the difference between BPTT, truncated BPTT and ARTBP.

Simply sampling arbitrarily long truncation lengths does not provide unbiasedness. Intuitively, it still favors short term gradient terms over long term ones. When using full BPTT, gradient computations flow back [3] from every timestep $t$ to every timestep $t' < t$. In truncated BPTT, gradients do not

---

[2] Usually the internal state $s_t$ is maintained from one subsequence to the other, not reset to a default value.

[3] Gradient flows between timesteps $t$ and $t'$ if there are no truncations occuring between $t$ and $t'$.

flow from $t$ to $t'$ if $t - t'$ exceeds the truncation length. In ARTBP, since random truncations are introduced, gradient computations flow from $t$ to $t'$ with a certain probability, decreasing with $t - t'$. To restore balance, ARTBP rescales gradient flows by their inverse probability. Informally, if a flow has a probability $p$ to occur, multiplication of the flow by $\frac{1}{p}$ restores balance on average.

Formally, at each training epoch, ARTBP starts by sampling a random sequence of truncation points, that is $(X_t)_{1 \leq t \leq T} \in \{0, 1\}^T$. A truncation will occur at all points $t$ such that $X_t = 1$. Here $X_t$ may have a probability law that depends on $X_1, \dots, X_{t-1}$, and also on the sequence of states $(s_t)_{1 \leq t \leq T}$ of the system. The reweighting factors that ARTBP introduces in the backpropagation equation depend on these truncation probabilities. (Unbiasedness is not obtained just by global importance reweighting between the various truncated subsequences: indeed, the backpropagation equation inside each subsequence has to be modified at every time step, see (11).)

The question of how to choose good probability distributions for the truncation points $X_t$ is postponed till Section 4. Actually, unbiasedness holds for any choice of truncation probabilities (Prop 1), but different choices for $X_t$ lead to different variances for the resulting gradient estimates.

**Proposition 1.** *Let $(X_t)_{t=1\dots T}$ be any sequence of binary random variables, chosen according to probabilities*

$$c_t := \mathbb{P}(X_t = 1 \mid X_{t-1}, \dots, X_1) \tag{10}$$

*and assume $c_t \neq 1$ for all $t$.*

*Define ARTBP to be backpropagation through time with a truncation between $t$ and $t + 1$ iff $X_t = 1$, and a compensation factor $\frac{1}{1-c_t}$ when $X_t = 0$, namely:*

$$\delta\tilde{\ell}_t := \begin{cases} \dfrac{\partial\ell}{\partial s}(s_t, o_t^*) & \text{if } X_t = 1 \text{ or } t = T \\[2mm] \dfrac{1}{1 - c_t}\delta\tilde{\ell}_{t+1}\dfrac{\partial F}{\partial s}(x_{t+1}, s_t, \theta) + \dfrac{\partial\ell}{\partial s}(s_t, o_t^*) & \text{otherwise.} \end{cases} \tag{11}$$

*Let $\tilde{g}$ be the gradient estimate obtained by using $\delta\tilde{\ell}_t$ instead of $\delta\ell_t$ in ordinary BPTT (7), namely*

$$\tilde{g} := \sum_{t=1}^{T} \delta\tilde{\ell}_t \frac{\partial F}{\partial\theta}(x_t, s_{t-1}, \theta) \tag{12}$$

*Then, on average over the ARTBP truncations, this is an unbiased gradient estimate of the total loss:*

$$\mathbb{E}_{X_1, \dots, X_T}[\tilde{g}] = \frac{\partial\mathcal{L}_T}{\partial\theta}. \tag{13}$$

The core of the proof is as follows: With probability $c_t$ (truncation), $\delta\tilde{\ell}_{t+1}$ does not contribute to $\delta\tilde{\ell}_t$. With probability $1 - c_t$ (no truncation), it contributes with a factor $\frac{1}{1-c_t}$. So on average, $\delta\tilde{\ell}_{t+1}$ contributes to $\delta\tilde{\ell}_t$ with a factor 1, and ARTBP (11) reduces to standard, non-truncated BPTT (8) on average. The detailed proof is given in Section 8.

While the ARTBP gradient estimate above is unbiased, some noise is introduced due to stochasticity of the truncation points. It turns out that ARTBP trades off memory consumption (larger truncation lengths) for variance, as we now discuss.

## 4 Choice of $c_t$ and memory/variance tradeoff

ARTBP requires specifying the probability $c_t$ of truncating at time $t$ given previous truncations. Intuitively the $c$'s regulate the average truncation lengths. For instance, with a constant $c_t \equiv c$, the lengths of the subsequences between two truncations follow a geometric distribution, with average truncation length $\frac{1}{c}$. Truncated BPTT with fixed truncation length $L$ and ARTBP with fixed $c = \frac{1}{L}$ are thus comparable memorywise.

Small values of $c_t$ will lead to long subsequences and gradients closer to the exact value, while large values will lead to shorter subsequences but larger compensation factors $\frac{1}{1-c_t}$ and noisier estimates.

In particular, the product of the $\frac{1}{1-c_t}$ factors inside a subsequence can grow quickly. For instance, a constant $c_t$ leads to exponential growth of the cumulated $\frac{1}{1-c_t}$ factors when iterating (11).

To mitigate this effect, we suggest to set $c_t$ to values such that the probability to have a subsequence of length $L$ decreases like $L^{-\alpha}$. The variance of the lengths of the subsequences will be finite if $\alpha > 3$. Moreover we might want to control the average truncation length $L_0$. This is achieved via

$$c_t = \mathbb{P}(X_t = 1 \mid X_{t-1}, \ldots, X_1) = \frac{\alpha - 1}{(\alpha - 2)L_0 + \delta t} \qquad (14)$$

where $\delta t$ is the time elapsed since the last truncation, $\delta t = t - \sup\{s \mid s < t, X_s = 1\}$. Intuitively, the more time spent without truncating, the lower the probability to truncate. This formula is chosen such that the average truncation length is approximately $L_0$, and the standard deviation from this average length is finite. The parameter $\alpha$ controls the regularity of the distribution of truncation lengths: all moments lower than $\alpha - 1$ are finite, the others are infinite. With larger $\alpha$, large lengths will be less frequent, but the compensating factors $\frac{1}{1-c_t}$ will be larger.

With this choice of $c_t$, the product of the $\frac{1}{1-c_t}$ factors incurred by backpropagation inside each subsequence grows polynomially like $L^{\alpha-1}$ in a subsequence of length $L$. If the dynamical system has geometrically decaying memory, i.e., if the operator norm of the transition operator $\frac{\partial F}{\partial s}$ is less than $1 - \varepsilon$ most of the time, then the value of $\delta \tilde{\ell}_t$ will stay controlled, since $(1 - \varepsilon)^L \cdot L^{\alpha}$ stays bounded. On the other hand, using a constant $c_t \equiv c$ provides bounded $\delta \tilde{\ell}_t$ only for small values $c < \varepsilon$.

In the experiments below, we use the $c_t$ from (14) with $\alpha = 4$ or $\alpha = 6$.

## 5 Online implementation

Importantly, ARTBP can be directly applied online, thus providing unbiased gradient estimates for recurrent networks.

Indeed, not all truncation points have to be drawn in advance: ARTBP can be applied by sampling the first truncation point, performing both forward and backward passes of BPTT up until this point, and applying a partial gradient descent update based on the resulting gradient on this subsequence. Then one moves to the next subsequence and the next truncation point, etc. (Fig. 1c).

## 6 Experimental validation

The experimental setup below aims both at illustrating the theoretical properties of ARTBP compared to truncated BPTT, and at testing the soundness of ARTBP on real world data.

### 6.1 Influence balancing

The influence balancing experiment is a synthetic example demonstrating, in a very simple model, the importance of being unbiased. Intuitively, a parameter has a positive short term influence, but a negative long term one that surpasses the short term effect. Practically, we consider a row of agents, numbered from left to right from 1 to $p + n$ who, at each time step, are provided with a signal depending on the parameter, and diffuse part of their current state to the agent directly to their left. The $p$ leftmost agents receive a positive signal at each time step, and the $n$ rightmost agents a negative signal. The training goal is to control the state of the leftmost agent. The first $p$ agents contribute positively to the first agent state, while the next $n$ contribute negatively. However, agent 1 only feels the contribution from agent $k$ after $k$ timesteps. If optimization is blind to dependencies above $k$, the effect of $k$ is never felt. A typical instantiation of such a problem would be that of a drug whose effect varies after various delays; the parameter to be optimized is the quantity of drug to be used daily.

Such a model can be formalized as Tallec & Ollivier (2017)

$$s_{t+1} = A\, s_t + (\theta, \ldots, \theta, -\theta, \ldots, -\theta)^{\top} \qquad (15)$$

with $A$ a square matrix of size $p + n$ with $A_{k,k} = 1/2$, $A_{k,k+1} = 1/2$, and 0 elsewhere; $s_t^k$ corresponds to the state of the $k$-th agent. $\theta \in \mathbb{R}$ is a scalar parameter corresponding to the intensity

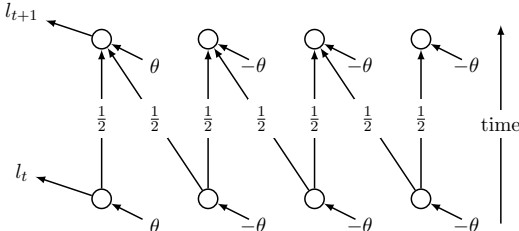

Figure 2: Influence balancing dynamics, 1 positive influence, 3 negative influences.

of the signal observed at each time step. The right-hand-side has $p$ positive-$\theta$ entries and $n$ negative-$\theta$ entries. The loss considered is an arbitrary target on the leftmost agent $s^1$,

$$\ell_t = \tfrac{1}{2}(s_t^1 - 1)^2. \tag{16}$$

The dynamics is illustrated schematically in Figure 2.

Fixed-truncation BPTT is experimentally compared with ARTBP for this problem. The setting is online: starting at $t = 1$, a first truncation length $L$ is selected (fixed for BPTT, variable for ARTBP), forward and backward passes are performed on the subsequence $t = 1, \ldots, L$, a vanilla gradient step is performed with the resulting gradient estimate, then the procedure is repeated with the next subsequence starting at $t = L + 1$, etc..

Our experiment uses $p = 10$ and $n = 13$, so that after 23 steps the signal should have had time to travel through the network. Truncated BPTT is tested with various truncations $L = 10, 100, 200$. (As the initial $\theta$ is fixed, truncated BPTT is deterministic in this experiment, thus we only provide a single run for each $L$.) ARTBP is tested with the probabilities (14) using $L_0 = 16$ (average truncation length) and $\alpha = 6$. ARTBP is stochastic: five random runs are provided to test reliability of convergence.

The results are displayed in Fig. 3. We used decreasing learning rates $\eta_t = \frac{\eta_0}{\sqrt{1+t}}$ where $\eta_0 = 3 \times 10^{-4}$ is the initial learning rate and $t$ is the timestep. We plot the average loss over timesteps 1 to $t$, as a function of $t$.

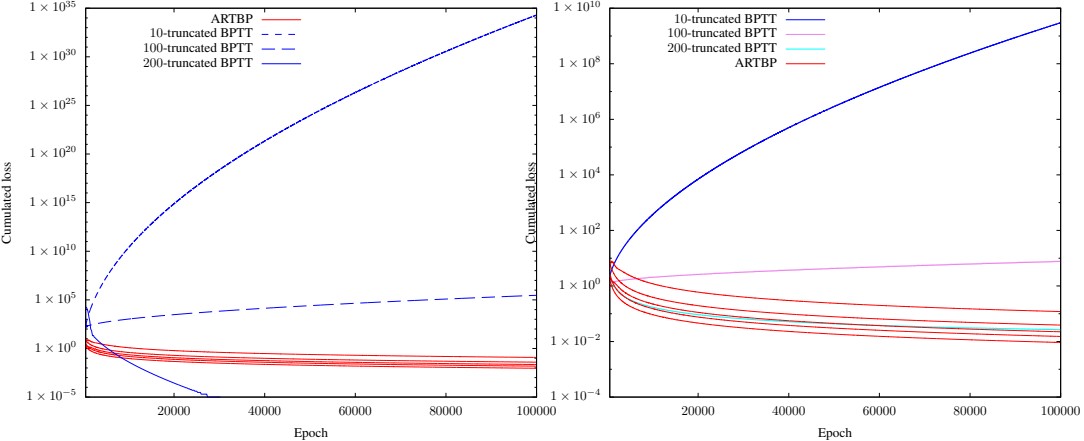

Figure 3: ARTBP and truncated BPTT on influence balancing, $n = 13$, $p = 10$. Note the log scale on the $y$-axis.

Truncated BPTT diverges even for truncation ranges largely above the intrinsic temporal scale of the system. This is an expected result: due to bias, truncated BPTT ill-balances temporal dependencies and estimates the overall gradient with a wrong sign. In particular, reducing the learning rate will *not* prevent divergence. On the other hand, ARTBP reliably converges on every run.

Note that for the largest truncation $L = 200$, truncated BPTT finally converges, and does so at a faster rate than ARTBP. This is because this particular problem is deterministic, so that a deterministic

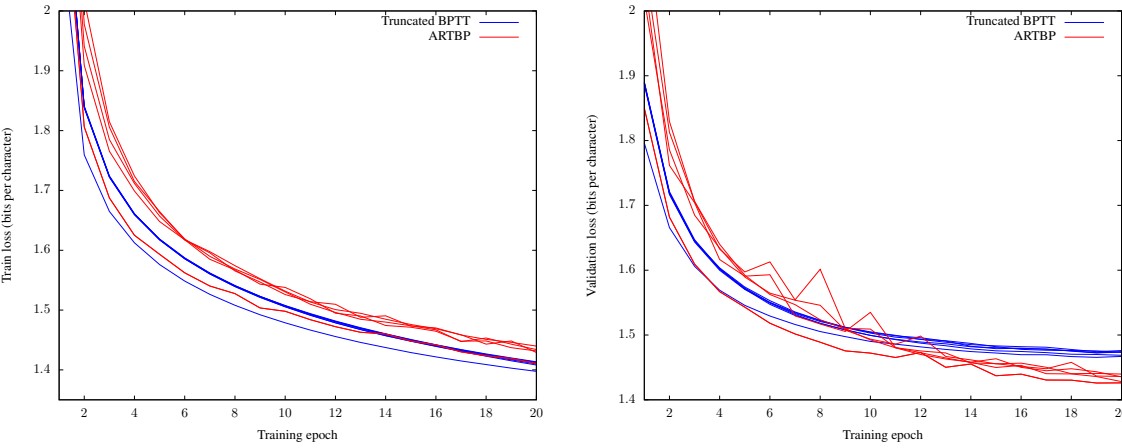

(a) Learning curves on Penn Treebank train set.      (b) Learning curves on Penn Treebank validation set.

Figure 4: Results on Penn Treebank character-level language modelling.

gradient scheme will converge (if it does converge) geometrically like $O(e^{-\lambda t})$, whereas ARTBP is stochastic due to randomization of truncations, and so will not converge faster than $O(t^{-1/2})$. This difference would disappear, for instance, with noisy targets or a noisy system.

**Character-level Penn Treebank language model.** We compare ARTBP to truncated BPTT on the character-level version of the Penn Treebank dataset, a standard set of case-insensitive, punctuation-free English text Marcus et al. (1993). Character-level language modelling is a common benchmark for recurrent models.

The dataset is split into training, validation and test sets following Mikolov et al. (2012). Both ARTBP and truncated BPTT are used to train an LSTM model Hochreiter & Schmidhuber (1997) with a softmax classifier on its hidden state, on the character prediction task. The training set is batched into $64$ subsets processed in parallel to increase computing speed. Before each full pass on the training set, the batched training sequences are split into subsequences:

- for truncated BPTT, of fixed size $50$;
- for ARTBP, at random following the scheme (14) with $\alpha = 4$ and $L_0 = 50$.

Truncated BPTT and ARTBP process these subsequences sequentially, [4] as in Fig. 1. The parameter is updated after each subsequence, using the Adam Kingma & Ba (2014) stochastic gradient scheme, with learning rate $10^{-4}$. The biases of the LSTM unit forget gates are set to 2, to prevent early vanishing gradients Gers et al. (2000). Results (in bits per character, bpc) are displayed in Fig. 4. Six randomly sampled runs are plotted, to test reliability.

In this test, ARTBP slightly outperforms truncated BPTT in terms of validation and test error, while the reverse is true for the training error (Fig. 4).

Even with ordinary truncated BPTT, we could not reproduce reported state of the art results, and do somewhat worse. We reach a test error of $1.43$ bpc with standard truncated BPTT and $1.40$ bpc with ARTBP, while reported values with similar LSTM models range from $1.38$ bpc Cooijmans et al. (2016) to $1.26$ bpc Graves (2013) (the latter with a different test/train split). This may be due to differences in the experimental setup: we have applied truncated BPTT without subsequence shuffling or gradient clipping Graves (2013) (incidentally, both would break unbiasedness). Arguably, the numerical issues solved by gradient clipping are model specific, not algorithm specific, while the point here was to compare ARTBP to truncated BPTT for a given model.

---

[4] Subsequences are not shuffled, as we do not reset the internal state of the network between subsequences.

## 7 CONCLUSION

We have shown that the bias introduced by truncation in the backpropagation through time algorithm can be compensated by the simple mathematical trick of randomizing the truncation points and introducing compensation factors in the backpropagation equation. The algorithm is experimentally viable, and provides proper balancing of the effects of different time scales when training recurrent models.

## 8 PROOF OF PROPOSITION 1

First, by backward induction, we show that for all $t \leq T$, for all $x_1, \ldots, x_{t-1} \in \{0, 1\}$,

$$\mathbb{E}\left[\delta\tilde{\ell}_t \mid X_{1:t-1} = x_{1:t-1}\right] = \delta\ell_t \tag{17}$$

where $\delta\ell_t$ is the value obtained by ordinary BPTT (8). Here $x_{1:k}$ is short for $(x_1, \ldots, x_k)$.

For $t = T$, this holds by definition: $\delta\tilde{\ell}_T = \frac{\partial\ell}{\partial s}(s_T, o_T^*) = \delta\ell_T$.

Assume that the induction hypothesis (17) holds at time $t + 1$. Note that the values $s_t$ do not depend on the random variables $X_t$, as they are computed during the forward pass of the algorithm. In particular, the various derivatives of $F$ and $\ell$ in (11) do not depend on $X_{1:T}$.

Thus

$$\mathbb{E}\left[\delta\tilde{\ell}_t \mid X_{1:t-1} = x_{1:t-1}\right] =$$

$$\mathbb{P}(X_t = 1 \mid X_{1:t-1} = x_{1:t-1})\,\mathbb{E}\left[\delta\tilde{\ell}_t \mid X_{1:t-1} = x_{1:t-1}, X_t = 1\right] + \tag{18}$$

$$\mathbb{P}(X_t = 0 \mid X_{1:t-1} = x_{1:t-1})\,\mathbb{E}\left[\delta\tilde{\ell}_t \mid X_{1:t-1} = x_{1:t-1}, X_t = 0\right] \tag{19}$$

$$= c_t\,\mathbb{E}\left[\delta\tilde{\ell}_t \mid X_{1:t-1} = x_{1:t-1}, X_t = 1\right] + (1 - c_t)\,\mathbb{E}\left[\delta\tilde{\ell}_t \mid X_{1:t-1} = x_{1:t-1}, X_t = 0\right] \tag{20}$$

If $X_t = 1$ then $\delta\tilde{\ell}_t = \frac{\partial\ell}{\partial s}(s_t, o_t^*)$. If $X_t = 0$, then $\delta\tilde{\ell}_t = \frac{\partial\ell}{\partial s}(s_t, o_t^*) + \frac{1}{1-c_t}\delta\tilde{\ell}_{t+1}\frac{\partial F}{\partial s}(x_{t+1}, s_t, \theta)$. Therefore, substituting into (20),

$$\mathbb{E}\left[\delta\tilde{\ell}_t \mid X_{1:t-1} = x_{1:t-1}\right] = \frac{\partial\ell}{\partial s}(s_t, o_t^*) + \mathbb{E}\left[\delta\tilde{\ell}_{t+1} \mid X_{1:t-1} = x_{1:t-1}, X_t = 0\right]\frac{\partial F}{\partial s}(x_{t+1}, s_t, \theta) \tag{21}$$

but by the induction hypothesis at time $t + 1$, this is exactly $\frac{\partial\ell}{\partial s}(s_t, o_t^*) + \delta\ell_{t+1}\frac{\partial F}{\partial s}(x_{t+1}, s_t, \theta)$, which is $\delta\ell_t$.

Therefore, $\mathbb{E}\left[\delta\tilde{\ell}_t\right] = \delta\ell_t$ unconditionally. Plugging the $\delta\tilde{\ell}$'s into (7), and averaging

$$\mathbb{E}_{X_1, \ldots, X_T}[\tilde{g}] = \sum_{t=1}^{T}\mathbb{E}_{X_t, \ldots, X_T}\left[\delta\tilde{\ell}_t\right]\frac{\partial F}{\partial\theta}(x_t, s_{t-1}, \theta) \tag{22}$$

$$= \sum_{t=1}^{T}\delta\ell_t\frac{\partial F}{\partial\theta}(x_t, s_{t-1}, \theta) \tag{23}$$

$$= \frac{\partial\mathcal{L}_T}{\partial\theta} \tag{24}$$

which ends the proof.

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
