# OpenReview forum: "Unbiasing Truncated Backpropagation Through Time"
_ICLR.cc/2018/Conference — Reject_

### Official Review · AnonReviewer2 · 2017-11-26
**Sufficient contribution, though little bad written**

**Rating:** 6
**Confidence:** 3

**Review:**

This paper proposes stochastic determination methods for truncation points in backpropagation through time. The previous truncation methods naively determine truncation points with fixed intervals, however, these methods cannot ensure the unbiasedness of gradients. The proposed methods stochastically determine truncation points with importance sampling. This framework ensures the unbiasedness of gradients, which contribute to the reliable convergence. Moreover, this paper investigates how the proposed methods work effectively by carefully tuning the sampling probability. This paper shows two experimental results, in which one is a simple synthetic task and the other is a real-data task. These results validate the effectiveness of the proposed methods.

Overall, I think the constitution and the novelty of this paper are above the bar. The proposed methods are simple extensions of the Truncated BPTT to ensure the unbiasedness. In particular, the investigation on the choice of the sampling probability is very helpful to consider how to enhance benefits of the proposed truncated BPTT methods. However, the written quality of this paper is not good at some points. I think the authors should re-check the manuscript and modify mistakes before the publication.

---

### Official Review · AnonReviewer1 · 2017-11-27

**Rating:** 5
**Confidence:** 4

**Review:**

This paper introduces a new approximation to backpropagation through time (BPTT) to overcome the computational and memory load that arise when having to learn from long sequences.
Rather than chopping the sequence into subsequences of equal length as in truncated BPTT, the authors suggest to segment the sequence into subsequences of differing lengths according to an a priori specified distribution for the segment length. The gradient estimator is made unbiased through a weighting procedure.

Whilst the proposed method is interesting and relevant, I find the analysis quite superficial and limited.

1) The distribution for the segment length is fully specified a priori. Depending on the problem at hand, different specifications could give rise to very different results. It would be good to suggest an approach for more automatically determine the (parameters of the) distribution.

2) Whilst unbiased, the proposed estimator could have high variance. This point is not investigated in the experimental section.

3) For an experimental paper as this one, it would be good to have many more problems analysed and a deeper analysis than the one given for the language problem.

---

### Official Review · AnonReviewer3 · 2017-11-27
**Interesting paper, would like to see more experiments**

**Rating:** 5
**Confidence:** 4

**Review:**

This is an interesting paper.

It is well known that TBPTT is biased because of a fixed truncation length. The authors propose to make it  unbiased by sampling different truncation lengths and hence changing  the optimization procedure which corresponds to adding noise in the gradient estimates which leads to  unbiased gradients.

Pros:

- Its a well written and easy to follow paper.
- If I understand correctly, they are changing the optimization procedure so that the proposed approach is able to find a local minima, which was not possible by using truncated backpropagation through time.
- Its interesting to see in there PTB results that they get better validation score as compared to truncated BPTT.

Cons:

- Though the approach is interesting, the results are quite preliminary. And given the fact there results are worse than the LSTM baseline (1.40 v/s 1.38). The authors note that it might be because of they are applying without sub-sequence shuffling.

- I'm not convinced of the approach yet. The authors could do some large scale experiments on datasets like Text8 or speech modelling.


Few points

- If I'm correct that the proposed approach indeed changes the optimization procedure, than I'd like to know what the authors think about exposure bias issue. Its a well known[1, 2] that we can't sample from RNN's for more number of steps, than what we used for trained (difference b/w teacher forcing and free running RNN). I'd like to know how does there method perform in such a regime (where you sample for more number of steps than you have trained for)

- Another thing, I'd like to see is the results of this model as compared to truncated backpropagation when you increase the sequence length. For example, Lets say you are doing language modelling on PTB, how the result varies when you change the length of the input sequence. I'd like to see a graph where on X axis is the length of the input sequence and on the Y axis is the bpc score (for PTB) and how does it compare to truncated backpropagation through time.

-  PTB dataset has still not very long term dependencies, so I'm curious what the authors think about using there method for something like speech modelling or some large scale experiments.

- I'd expect the proposed approach to be more computationally expensive as compared to Truncated Back-propagation through time. I dont think the authors mentioned this somewhere in the paper. How much time does a single update takes as compared to Truncated Back-propagation through time ?

- Does the proposed approach help in flow of gradients?

- In practice for training RNN's people use gradient clipping which also makes the gradient biased. Can the proposed method be used for training longer sequences?

[1] Scheduled Sampling For Sequence Prediction with RNN's https://arxiv.org/abs/1506.03099
[2] Professor Forcing  https://arxiv.org/abs/1610.09038


Overall, Its an interesting paper which requires some more analysis to be published in this conference. I'd be very happy to increase my score if the authors can provide me results what I have asked for.

---

### Decision · Program_Chairs · 2018-01-29
**ICLR 2018 Conference Acceptance Decision**

**Decision:**

Reject

**Comment:**

Meta score: 5

The paper explores an interesting idea, addressing a known bias in truncated BPTT by sampling across different truncated history lengths.  Limited theoretical analysis is presented along with PTB language modelling experimentation.   The experimental part could be stronger (e.g. trying to improve over the baseline) and perhaps more than just PTB.

Pros:
 - interesting idea
Cons:
 - limited analysis
 - limited experimentation